# FedPHA: Federated Prompt Learning for Heterogeneous Client Adaptation

Chengying Fang [* 1]  Wenke Huang [* 1]  Guancheng Wan [* 1]  Yihao Yang [1]  Mang Ye [1]

## Abstract

Federated Prompt Learning (FPL) adapts pre-trained Vision-Language Models (VLMs) to federated learning through prompt tuning, leveraging their transferable representations and strong generalization capabilities. Traditional methods often require uniform prompt lengths for federated aggregation, limiting adaptability to clients with diverse prompt lengths and distribution biases. In this paper, we propose **Fed**erated **P**rompt Learning for **H**eterogeneous Client **A**daptation (FedPHA), a novel framework that combines a fixed-length global prompt for efficient aggregation with local prompts of varying lengths to capture client-specific data characteristics. Additionally, FedPHA designs Singular Value Decomposition (SVD) based projection and bidirectional alignment to disentangle global conflicts arising from client heterogeneity, ensuring that personalized client tasks effectively utilize non-harmful global knowledge. This approach ensures that global knowledge improves model generalization while local knowledge preserves local optimization. Experimental results validate the effectiveness of FedPHA in achieving a balance between global and personalized knowledge in federated learning scenarios. The source code is available at: https://github.com/CYFang6/FedPHA.

## 1. Introduction

Federated learning (McMahan et al., 2017; Yang et al., 2019; Hong & Chae, 2021; Qu et al., 2022; Huang et al., 2024), as a distributed machine learning paradigm, addresses data silos by enabling participants to collaboratively train models locally, ensuring data privacy while promoting AI collabora-

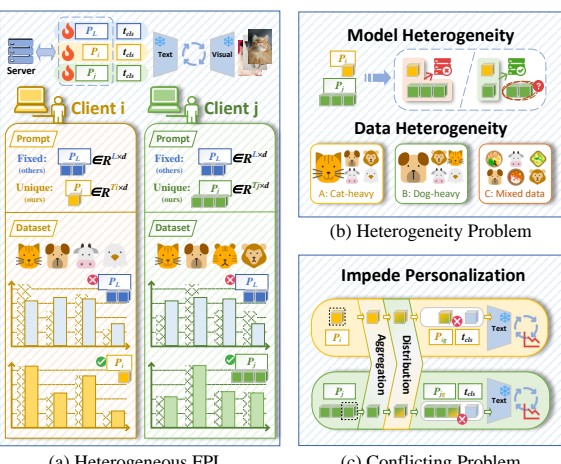

Figure 1: **Problem illustration** of heterogeneous federated prompt learning (FPL). (a) Heterogeneous FPL: Clients hold different prompts, models, and data distributions. (b) Heterogeneity Problem: Aggregation of heterogeneous prompts and models under non-IID data is inherently challenging. (c) Conflicting Problem: Aggregated global prompts may conflict with client-specific knowledge, impeding personalization during local adaptation.

tion. However, existing federated learning approaches face significant limitations due to the frequent exchange large volumes of model parameters with a central server. This results in high communication overhead, increased training costs, potential performance degradation, and instability during the training process (Wu et al., 2020; Kulkarni et al., 2020; Chen et al., 2022; Wan et al., 2024).

Fortunately, vision-language pre-trained models such as Contrastive Language-Image Pretraining (CLIP) (Radford et al., 2021) have demonstrated potential in learning robust and versatile representations suitable for various image distributions, aligning well with the objectives of federated learning. However, the substantial communication overhead between the server and clients poses challenges for training CLIP within federated learning frameworks (Lu et al., 2023). Additionally, overfitting concerns may arise when large-scale models are trained on limited client data. Prompt learning (Zhou et al., 2022b;a; Khattak et al., 2023; khattak et al., 2023; Li et al., 2024b) offers a flexible approach to adapt pre-trained models to downstream tasks by training only additional parameters. This enables prompts to capture task-specific information while guiding the performance of

[*]Equal contribution [1]National Engineering Research Center for Multimedia Software, School of Computer Science, Wuhan University, Wuhan, China. Correspondence to: Mang Ye <ye-mang@whu.edu.cn>.

*Proceedings of the 42^{st} International Conference on Machine Learning*, Vancouver, Canada. PMLR 267, 2025. Copyright 2025 by the author(s).

the fixed model. Leveraging its lightweight nature, prior research (Guo et al., 2023b; Qiu et al., 2023; Feng et al., 2023; Su et al., 2024) has explored integrating prompt learning into federated learning to address these challenges.

As shown in Figure 1, one of the fundamental challenges in federated learning is client heterogeneity (Li et al., 2021b;a; Huang et al., 2022; Wang et al., 2023; Huang et al., 2023b; Hu et al., 2024; Tan et al., 2025), which manifests in two key forms: data heterogeneity, where client data distributions are non-IID, and model heterogeneity, where clients employ diverse model architectures or have varying computational resources. These challenges significantly hinder model convergence and system efficiency. Intuitively, different clients should require prompts of varying lengths to more effectively capture the characteristics of their local data (proved in Sec 4.3). However, due to aggregation constraints, current federated prompt learning frameworks (Guo et al., 2023b;a; Feng et al., 2023; Yang et al., 2023; Bai et al., 2024) typically enforce uniform prompt lengths across all clients to facilitate the aggregation process. Although some works (Li et al., 2024a; Cui et al., 2024) have proposed dual-layer architectures incorporating both global and local prompts while aggregating only global prompts, the structural constraints of these methods prevent support for varying local prompt lengths. In such approaches, forcibly expanding or reducing the length of prompts may lead to information loss, further highlighting the challenge of designing methods that can adapt to different prompt length requirements.

Furthermore, the complex interplay between shared global knowledge and client-specific local knowledge presents another critical challenge in federated prompt learning, as illustrated in Figure 1(c). While global knowledge aggregated from multiple clients can provide valuable generalizable features, it may also contain potentially conflicting information that hinders local optimization (Wang et al., 2020; Li et al., 2022; Nguyen et al., 2024). Specifically, in high data heterogeneity scenarios, the knowledge learned from other clients may not align with or even contradict the optimal features required for a specific client's local task. The conventional federated averaging approach (Guo et al., 2023b;a) may force clients to compromise their locally optimal representations to accommodate the global consensus, potentially degrading performance on client-specific tasks. Although some recent works (Guo et al., 2023a; Li et al., 2024a) have attempted to mitigate this issue by separating global and local prompts, they lack explicit mechanisms to resolve knowledge conflicts and ensure effective knowledge transfer between the two.

To address the challenges in Figure 1, our work is motivated by two primary objectives: (1) Designing a framework that balances federated learning aggregation requirements with client-side flexibility, accommodating diverse prompt lengths and varying data distributions while preserving information integrity. (2) Developing a method to mitigate the negative impact of the conflicting parts between global and local knowledge, allowing clients to retain their unique characteristics while benefiting from global knowledge.

In this work, we propose *FedPHA* (**Fed**erated **P**rompt Learning for **H**eterogeneous Client **A**daptation), a novel method designed to address challenges related to data and model heterogeneity, as well as resolving conflicts between global and local knowledge. FedPHA designs a G-L (Global-Local) architecture to manage the varying prompt requirements of heterogeneous clients. Each client receives a local prompt with a unique length and a global prompt with a uniform length. These prompts are connected through shared tokens and a frozen encoder, establishing an implicit coupling between global and local prompts. To resolve conflicts between global and local knowledge, we integrate an SVD-based projection mechanism, which filters out conflicting parts while preserving essential local information. In addition, we introduce a bidirectional alignment function in the optimization process. This ensures alignment between local and projected features while ensuring a clear distinction between global and local features, preserving their unique characteristics. Our main contributions are summarized as:

- We are the first to consider the heterogeneity of prompt lengths in federated prompt learning. We design a G-L framework to facilitate aggregation and individual client requirements. Shared tokens and a frozen encoder connect the global and local prompts, creating implicit coupling. We use feature-level computation to prevent information loss from prompt length variations.

- We devise an SVD-based projection mechanism to disentangle conflicting parts between global and local knowledge, retaining essential local information while removing inconsistencies. And bidirectional alignment function aligns local and projected features and preserves the unique characteristics of global and local representations.

- We evaluate FedPHA against the existing personalized techniques on widely-adopted datasets. Extensive experiments and ablation studies demonstrate the superiority of our methods under heterogeneous settings.

## 2. Related Work

### 2.1. Heterogeneous Federated Learning

Federated Learning (FL) aims to address the critical challenge of heterogeneity in client data distributions (Xu et al., 2021; Huang et al., 2022; Fang & Ye, 2022; Huang et al., 2023a). Key types of heterogeneity include label shift, where the label distribution $P(Y)$ differs across clients while $P(X|Y)$ remains consistent, and domain shift, where the feature distribution $P(X)$ varies while $P(Y)$ stays un-

changed. These challenges necessitate specialized methods to ensure effective collaboration across diverse client datasets. To handle such heterogeneity, early approaches in FL often include incorporated regularization terms to the loss function (Li et al., 2020) or fine-tuning the global model on clients' local datasets (Fallah et al., 2020). However, these methods risk local overfitting due to the limited and diverse data on clients, potentially compromising global generalizability. More advanced methods explicitly aim to balance global and local models (Chen & Chao, 2022), or leverage client relationships through weighted aggregation techniques, such as FedPAC (Xu et al., 2023) and FedDisco (Ye et al., 2023). Parameter decomposition has been explored for heterogeneity; e.g., FedTP (Li et al., 2023) learns client-specific self-attention layers. Despite progress, FL methods continue to struggle with balancing personalization and generalization under high data and model heterogeneity. Our proposed FedPHA addresses these challenges by improving the balance between global consistency and local personalization under prompt-length heterogeneity.

## 2.2. Federated Prompt Learning

Prompt learning, initially developed for NLP, has been extended to Vision-Language Models to adapt pre-trained models to diverse downstream tasks. Early methods like CLIP (Radford et al., 2021) used manual templates, while newer approaches learn prompts in continuous embedding spaces. For example, CoOp (Zhou et al., 2022b) fine-tunes CLIP with continuous vectors, and ProGrad (Zhu et al., 2023) selectively updates prompts to preserve essential VLM knowledge. To integrate prompt learning into Federated Learning (FL), methods like FedPrompt (Zhao et al., 2023) and PromptFL (Guo et al., 2023b) accelerate global aggregation and address limited user data. Building on these, pFedprompt (Guo et al., 2023a) employs a non-parametric personalized attention module for local feature generation, and pFedPG (Yang et al., 2023) designs a server-side prompt generator for client-specific personalization. FedOTP (Li et al., 2024a) uses unbalanced Optimal Transport to coordinate global and local prompts. FedPGP (Cui et al., 2024) adapts to heterogeneous data via low-rank decomposition of global prompts and contrastive loss to balance personalization and generalization. However, the structural limitations of these methods prevent them from accommodating varying local prompt lengths and lack a mechanism to separate conflicting global knowledge from personalized local knowledge. In contrast, our proposed FedPHA leverages a dual-layer architecture and Singular Value Decomposition (SVD) to effectively address these challenges.

## 2.3. Singular Value Decomposition

Singular Value Decomposition (SVD) (Golub et al., 1987) is a technique that decomposes a matrix $A \in \mathbb{R}^{m \times n}$ into

$A = USV^\top$, where $U \in \mathbb{R}^{m \times m}$, $V \in \mathbb{R}^{n \times n}$ are orthonormal matrices, and $S \in \mathbb{R}^{m \times n}$ is a diagonal matrix of singular values. SVD enables dimensionality reduction by retaining only the largest singular values. SVD has been widely explored in large language models (LLMs) for its ability to decompose matrices into orthogonal components, offering robust mathematical foundations for a variety of applications. It is a powerful tool for dimensionality reduction (Hsu et al., 2022; Yuan et al., 2023; Saha et al., 2023; Wang et al., 2024), enabling the extraction of key features while minimizing redundant information. Additionally, SVD has proven effective in noise filtering (Sharma et al., 2023; Dai et al., 2024), as it isolates signal-dominant components and suppresses less significant, noisy contributions. Furthermore, it is frequently utilized in subspace projection (Feng et al., 2023; Lan et al., 2024), enabling data representation in lower-dimensional subspaces while preserving essential properties and optimizing computational efficiency. Our work builds on these principles by proposing an SVD-based projection mechanism in the context of federated prompt learning, addressing heterogeneity and reducing potential conflicts between local and global information.

## 3. Proposed Method

In this section, we present the details of FedPHA illustrated in Figure 2. To address the issue that existing methods cannot adapt to heterogeneous prompt lengths, FedPHA introduces a G-L heterogeneous federated prompt architecture (Sec 3.2). Meanwhile, to reduce the negative impact caused by the conflict between global prompts and local prompts, we propose SVD-based projection (Sec 3.3) and bidirectional alignment (Sec 3.4). The details of our FedPHA are provided in Algorithm 1.

### 3.1. Preliminaries of Prompt Learning

Prompt learning efficiently adapts pre-trained models like CLIP for downstream tasks by introducing learnable parameters in the text encoder. Unlike zero-shot transfer, which uses fixed word embeddings $W = \{w_1, w_2, \ldots, w_l\}$ from handcrafted prompts (e.g., "a photo of a $\langle$label$\rangle$"), prompt learning adds learnable continuous context vectors $P_t = \{p_1, p_2, \ldots, p_T\} \in \mathbb{R}^{T \times d}$, where $T$ is the prompt length and $d$ is the embedding dimension. This allows the text encoder to capture task-specific information while keeping the image encoder fixed. For a class label $t_{\text{Class}}$, the textual input is extended as:

$$\tilde{Y}_p = \{t_{\text{SOS}}, P_t, t_1, t_2, \ldots, t_L, t_{\text{Class}}, t_{\text{EOS}}\}, \quad (1)$$

where $t_{\text{SOS}}$ and $t_{\text{EOS}}$ are learnable start/end embeddings, $\{t_1, \ldots, t_L\}$ are fixed word embeddings, and $t_{\text{Class}}$ is the class label embedding. The text encoder $g(\cdot)$, composed of transformer layers, generates the prompted textual feature

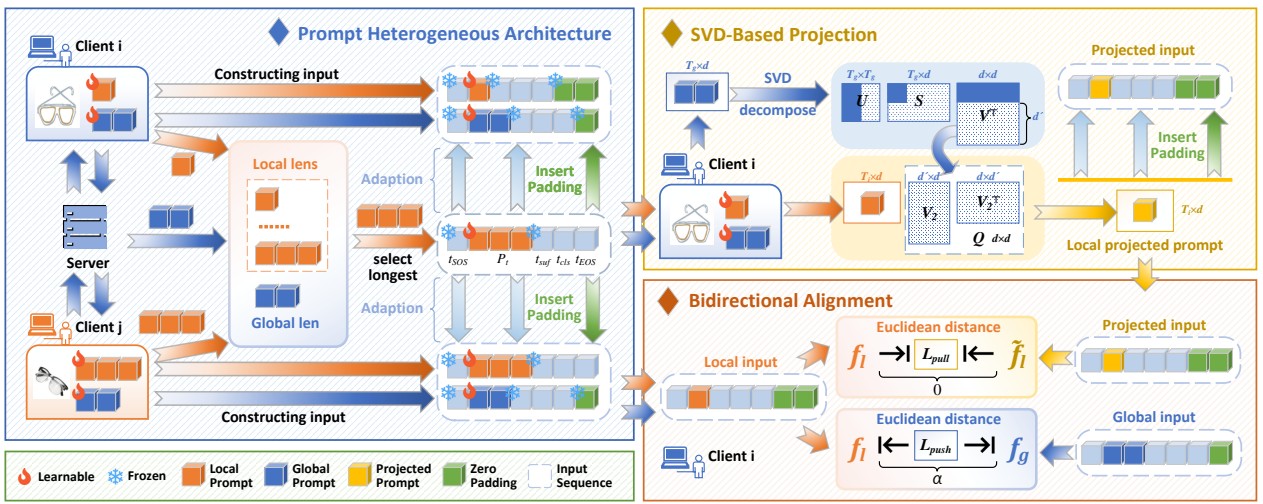

Figure 2: Our proposed FedPHA framework for heterogeneous federated prompt learning consists of three main components. The dual-layer architecture (Sec 3.2) assigns each client a local prompt of varying lengths and a global prompt of uniform length. The longest prompt forms frozen tokens ($t_{SOS}$, $t_{Class}$, $t_{Suffix}$, $t_{EOS}$), distributed to all clients for input sequence construction, with $t_{Padding}$ filling gaps. The top right shows the SVD-Based Projection (Sec 3.3), where the global prompt is decomposed via SVD. The last $d'$ components of $V^T$ form the null space, into which local prompts are projected. The bottom right illustrates the bidirectional alignment (Sec 3.4), which uses $\mathcal{L}_{pull}$ to align local and projected features, and $\mathcal{L}_{push}$ to separate local and global features. Only local features are used for final inference.

$\tilde{g}_p = g(\tilde{Y}_p, \theta_g) \in \mathbb{R}^d$, where $\theta_g$ includes frozen pre-trained parameters and the learnable $P_t$. For image classification, textual features $\{\tilde{g}_{pi}\}_{i=1}^{C}$ are compared with image features $\tilde{f}$, extracted by the frozen image encoder $f(\cdot)$. The class $k$ probability for an image $x$ is:

$$P(\hat{y} = k|x) = \frac{\exp(\text{sim}(\tilde{g}_{pk}, \tilde{f})/\tau)}{\sum_{i=1}^{C} \exp(\text{sim}(\tilde{g}_{pi}, \tilde{f})/\tau)}, \quad (2)$$

where $\text{sim}(\cdot, \cdot)$ is cosine similarity, and $\tau$ regulates the Softmax sharpness. The learnable prompts $P_t$ are optimized with cross-entropy loss. For a dataset $\mathcal{D}$ of input-output pairs $(X, y)$, the objective is:

$$\mathcal{L}_{\text{CE}} = \arg\min_{P_t} \mathbb{E}_{(X,y)\sim\mathcal{D}} \mathcal{L}(\text{sim}(\tilde{g}_p, \tilde{f}), y). \quad (3)$$

### 3.2. Federated Prompt Heterogeneous Architecture

In federated learning, different clients often exhibit diverse data distributions and task requirements, making it difficult to use a single, fixed-length prompt that balances personalization and aggregability. To address this issue, we propose a G-L heterogeneous federated prompt architecture, which comprises a fixed-length global prompt and a variable-length local prompt for each client. Additionally, we employ frozen contextual tokens and zero-padding tokens to ensure the implicit coupling between global and local prompts, providing a consistent structure that facilitates subsequent feature computation.

Suppose there are $N$ clients indexed by $i = 1, 2, \ldots, N$ and a central server. Each client $i$ holds a local dataset $\mathcal{D}_i$ of size $n_i$, with $\{\mathcal{D}_1, \ldots, \mathcal{D}_N\}$ collectively denoting the full dataset. Let $\mathcal{C}_r \subseteq \{1, \ldots, N\}$ be the subset of clients selected at communication round $r$. Each selected client performs local training for $E$ epochs using a loss $\mathcal{L}$. During local optimization, the global prompt $P_{g,i}^{r,e} \in \mathbb{R}^{T_g \times d}$ and the local prompt $P_{l,i}^{r,e} \in \mathbb{R}^{T_i \times d}$ (where $T_g$ is fixed and $T_i$ may vary across clients) are updated. Let $\eta$ be the learning rate. For each local epoch $e$, the updates are

$$P_{*,i}^{r,e+1} = P_{*,i}^{r,e} - \eta \nabla_{P_{*,i}} \mathcal{L}\left(P_{g,i}^{r,e}, P_{l,i}^{r,e}; \mathcal{D}_i\right), \quad (4)$$

where $P_{*,i}^{r,e}$ denotes either the global prompt $P_{g,i}^{r,e}$ or the local prompt $P_{l,i}^{r,e}$.

At the end of local training, only $P_{g,i}^{r,E}$ is uploaded to the server, while $P_{l,i}^{r,E}$ is kept locally to preserve personalized information. The server then aggregates the global prompts from all clients $i \in \mathcal{C}_r$. The new global prompt is obtained by weighted averaging based on the sample sizes $n_i$ of the participating clients:

$$P_g^{(r+1,0)} = \sum_{i \in \mathcal{C}_r} \frac{n_i}{\sum_{j \in \mathcal{C}_r} n_j} P_{g,i}^{r,E}, \quad (5)$$

where $n_i$ denotes the number of samples in the local dataset $\mathcal{D}_i$. This ensures that clients with larger datasets contribute more significantly to the aggregated global prompt. The updated global prompt $P_g^{(r+1,0)}$ is then distributed to all

clients in preparation for the next round of communication. Formally, the objective function can be expressed as:

$$\min_{P_g, \{P_{l,i}\}_{i=1}^N} \quad \sum_{i=1}^N \frac{n_i}{\sum_{j=1}^N n_j} \mathcal{L}_i\left(P_{g,i}^{r,e}, P_{l,i}^{r,e}; \mathcal{D}_i\right), \quad (6)$$

where $\mathcal{L}_i(P_{g,i}^{r,e}, P_{l,i}^{r,e}; \mathcal{D}_i)$ denotes the local loss of client $i$. During training, both global and local prompts are leveraged to jointly optimize the model. During inference, only local features, derived from the local prompts, are used for final inference to ensure adaptability to personalized data distributions.

**Heterogeneous Key.** The fundamental distinction of our method from others lies in its unique handling of global and local prompts. Specifically, on the text-encoder side, each client is assigned a local prompt of varying lengths and a global prompt of fixed length. We select the longest prompt among all clients to construct the frozen $t_{\text{SOS}}, t_{\text{Class}}, t_{\text{Suffix}}, t_{\text{EOS}}$ via Eq.(1). Each client then forms three types of input sequences by concatenating its prefix tokens, class name tokens, and suffix tokens with: (1) the global prompt $P_{g,i}^{r,e}$, (2) the local prompt $P_{l,i}^{r,e}$, and (3) the projected local prompt $\widetilde{P}_{l,i}^{r,e}$ (in Eq.(10)). If the resulting sequence is shorter than the maximum encoder length $L_{\max}$, zero vectors ($t_{\text{Padding}}$) are appended. The final input sequence can be represented as:

$$\tilde{Y}_p = \{t_{\text{SOS}}, P_t, t_{\text{Class}}, t_{\text{Suffix}}, t_{\text{EOS}}, t_{\text{Padding}}\}, \quad (7)$$

The global and local prompts are implicitly coupled and interact through shared prefix and suffix tokens, as well as a common Transformer encoder. This interaction ensures that while local prompts retain their client-specific distinctions, the overall model still operates within a shared high-dimensional representation space, promoting information exchange across clients. Finally, the resulting input sequences are fed into the pre-trained CLIP encoder alongside image representations to compute similarity scores and perform classification.

### 3.3. SVD-Based Projection

Although the above framework achieves personalization with heterogeneous prompt lengths by thoroughly separating global prompts and local prompts, an interaction mechanism between global and local prompts is still required to facilitate information exchange. Simple alignment or orthogonality between the two may be insufficient in cases of highly heterogeneous data distribution. Therefore, it is necessary to further refine *local prompts* to mitigate potential conflicts with *global prompts*. Inspired by subspace projection in matrix factorization, we propose a projection mechanism based on Singular Value Decomposition (SVD) to filter out unnecessary or conflicting components from local prompts.

If $P_g^{r,e} \in \mathbb{R}^{T_g \times d}$ be the current global prompt at local epoch $e$. We directly perform singular value decomposition (SVD) on $P_g^{r,e}$, obtaining

$$P_g^{r,e} = U S V^\top, \quad (8)$$

where $U \in \mathbb{R}^{T_g \times T_g}$ and $V \in \mathbb{R}^{d \times d}$ are orthonormal matrices, and $S \in \mathbb{R}^{T_g \times d}$ is a diagonal matrix with descending singular values. Because the global prompt has length $T_g$ and the local prompt may have a different length $T_i$, choosing $U$ to construct the projection could lead to dimension mismatch. Therefore, we utilize $V$, which naturally resides in the same feature dimension $d$ as both local and global prompts, to form the null space. Let $V_2 \in \mathbb{R}^{d \times d'}$ be the matrix collecting the columns of $V$ corresponding to the smaller singular values. The number of selected columns $d'$ is determined by the hyperparameter ratio $\rho$:

$$d' = \lfloor (1 - \rho)d \rfloor. \quad (9)$$

These directions typically capture less significant or potentially conflicting components in the global prompt. The null-space projection matrix $Q$ is then defined as $Q = V_2 V_2^\top$. Then the projection of the local prompt is defined as:

$$\widetilde{P}_{l,i}^{r,e} = P_{l,i}^{r,e} Q = P_{l,i}^{r,e} V_2 V_2^\top, \quad (10)$$

where $P_{l,i}^{r,e} \in \mathbb{R}^{T_i \times d}$ is the local prompt for client $i$. The projected prompt $\widetilde{P}_{l,i}^{r,e} \in \mathbb{R}^{T_i \times d}$ retains the same dimensions as the original local prompt. By projecting $P_{l,i}^{r,e}$ onto $Q$, we effectively "filter out" dimensions dominated by the global prompt's major components, thereby reducing potential conflicts between local and global information. This step is crucial in heterogeneous settings: it preserves local discriminative features relevant to each client's data while mitigating interference from global prompt directions that may not generalize to individual client distributions.

### 3.4. Bidirectional Alignment

To mitigate conflicts, we project the local prompt $P_{l,i}^{r,e}$ into the null space. However, this may lead to information loss, reducing client-specific expressiveness. To address this, we introduce a bidirectional alignment mechanism: a "pull" term to retain information by aligning the local prompt with its projection and a "push" term to ensure sufficient divergence from the global prompt.

To ensure that the local prompt does not deviate excessively from its projected prompt, we minimize the mean squared error (MSE) between $f(P_{l,i}^{r,e})$ and $f(\widetilde{P}_{l,i}^{r,e})$. Formally,

$$\mathcal{L}_{\text{pull}} = \left\| f(P_{l,i}^{r,e}) - f(\widetilde{P}_{l,i}^{r,e}) \right\|_2^2, \quad (11)$$

which encourages $P_{l,i}^{r,e}$ to "pull" closer to its null-space-projected version $\widetilde{P}_{l,i}^{r,e}$ in the feature space. By doing so, we

retain essential local information while filtering out components that conflict with the global prompt.

In parallel, to prevent the local prompt from collapsing too closely to the global prompt, we introduce a margin-based "push" term to maintain a safe distance between $f\left(P_{l,i}^{r,e}\right)$ and $f\left(P_g^{r,e}\right)$. Specifically,

$$\mathcal{L}_{\text{push}} = \text{ReLU}\left(\alpha - \|f(P_{l,i}^{r,e}) - f(P_g^{r,e})\|_2\right), \quad (12)$$

where $\alpha > 0$ defines the minimal acceptable distance. This ensures that each local prompt remains sufficiently personalized and does not become overly dominated by the global prompt's major components.

In practice, both of these MSE-based terms (*pull* and *push*) are combined with standard cross-entropy losses (for both the local prompt and the global prompt), forming a unified training objective:

$$\mathcal{L} = \mathcal{L}_{\text{CE}}(P_{l,i}^{r,e}, y_i) + \mathcal{L}_{\text{CE}}(P_g^{r,e}, y_i) + \mathcal{L}_{\text{pull}} + \mathcal{L}_{\text{push}}. \quad (13)$$

This bidirectional alignment strategy mitigates potential conflicts arising from heterogeneous data distributions, ensuring that local prompts retain discriminative characteristics while still benefiting from global knowledge.

## 4. Experiments

In this section, we conduct extensive experiments aiming at answer following research questions:

- **Q1:** Does the proposed method maintain its effectiveness when the prompt length is fixed? How does it compare to the state-of-the-art (SOTA) methods? (in Sec 4.2)
- **Q2:** For clients with diverse data distributions, can prompts of varying lengths enhance performance? Does length heterogeneity provide any advantage? (in Sec 4.3)

### 4.1. Experimental Setup

**Datasets.** Following previous research (Guo et al., 2023b;a), we evaluate our method on multiple public benchmark datasets exhibiting significant data heterogeneity. We use five visual classification datasets—Food101 (Bossard et al., 2014), DTD (Cimpoi et al., 2014), Caltech101 (Fei-Fei, 2004), Flowers102 (Nilsback & Zisserman, 2008), and OxfordPets (Parkhi et al., 2012)—collectively referred to as the CLIP dataset (1 domain). These datasets are configured using a pathological non-IID setting, where each client is randomly allocated a distinct number of non-overlapping classes to simulate heterogeneous data distributions. In addition, we select two cross-domain datasets, Office31 (Saenko et al., 2010) (3 domains) and OfficeHome (Venkateswara et al., 2017) (4 domains), where the data for each client is drawn from a specific domain, further emphasizing data heterogeneity. Finally, we employ two classic image benchmark datasets, CIFAR10 (Krizhevsky et al., 2010) and

---

**Algorithm 1 Overall Procedure of FedPHA**

**Data:** The random public dataset $\{\mathcal{D}_i\}_{i=1}^N$, sizes $\{n_i\}$;

**Input:** Communication rounds $R$; Local epochs $E$; Learning rate $\eta$; SVD ratio $\rho$; Margin $\alpha$; Initial global prompt $P_g^{(0)}$ Initial local prompts $P_{l,i}^{(0)}$.

**Output:** The final local models $M_i^R$

```
// Federated Rounds
for r = 1, 2, ..., R do
    // Participant Side
    for each client i ∈ C_r in parallel do
        P_{g,i}^{(r,E)}, P_{l,i}^{(r,E)} ← LocalUpdate(P_{g,i}^{(r,1)}, P_{l,i}^{(r-1,E)})
    end
    // Server Side
    P_g^{(r+1,1)} ← Σ_{i∈C_r} (n_i / Σ_{j∈C_r} n_j) P_{g,i}^{(r,E)} in Eq.(5)
end
// Local Epochs
for e = 1, 2, ..., E do
    // SVD-Based Projection
    P_{g,i}^{r,e} ← USV^⊤ in Eq.(8)

    V_2 ← V[:, d(1 − ρ) : d] via Eq.(5)

    Q ← V_2 V_2^⊤  // Construct projection
    P̃_{l,i}^{r,e} ← P_{l,i}^{r,e} Q in Eq.(10)

    // Compute losses
    L_pull ← ||f(P_{l,i}^{r,e}) − f(P̃_{l,i}^{r,e})||² in Eq.(11)

    L_push ← ReLU(α − ||f(P_{l,i}^{r,e}) − f(P_{g,i}^{r,e})||) in Eq.(12)

    L ← L_CE + L_pull + L_push in Eq.(13)

    // Update prompts using gradients
    Update P_{g,i}^{(r,e+1)}, P_{l,i}^{(r,e+1)} using ∇L
end
return M_i^R  // Final model after R rounds
```

---

CIFAR-100 (Krizhevsky & Hinton, 2009), where data is randomly partitioned among clients using a symmetric Dirichlet distribution as in (Cao et al., 2023; Shamsian et al., 2021) with $\beta = 0.5$, further enhancing the diversity in data distribution. Details of these dataset setups are provided in Appendix Section B.1.

**Baselines.** We compare our FedPHA with five baseline methods: (1) Zero-shot CLIP (Radford et al., 2021), a local training approach that utilizes manually designed text prompt templates to generate the model's initial performance. (2) PromptFL (Guo et al., 2023b), a prompt-based federated learning method that learns a unified prompt across clients using the federated averaging mechanism (McMahan et al., 2017). (3) PromptFL+Prox (Li et al., 2020), as introduced in (Guo et al., 2023a), which con-

Table 1: **Comparison with the SOTA methods with same prompt length on single-domain datasets** across 10 clients.

| Methods | Caltech101 | Food101 | Flowers102 | OxfordPets | DTD |
|---|---|---|---|---|---|
| Zero-Shot CLIP (Radford et al., 2021) | 91.43±0.24 | 85.43±0.05 | 67.70±0.12 | 88.95±0.12 | 43.28±0.10 |
| PromptFL (Guo et al., 2023b) | 93.47±0.42 | 86.60±0.10 | 85.54±0.65 | 93.44±0.25 | 55.24±0.31 |
| Prompt+Prox (Li et al., 2020) | 93.55±0.36 | 86.65±0.17 | 85.74±0.25 | 93.48±0.32 | 55.64±0.23 |
| FedPGP (Cui et al., 2024) | 95.86±0.50 | 88.30±0.41 | 94.46±2.44 | 93.87±0.31 | 62.95±3.01 |
| FedOTP (Li et al., 2024a) | 98.11±0.04 | 92.96±0.16 | 98.47±0.07 | 98.60±0.11 | 89.97±0.17 |
| FedPHA | **99.05±0.08** | **96.42±0.05** | **99.25±0.05** | **99.25±0.06** | **91.79±0.17** |

Table 2: **Comparison with the SOTA methods with the same prompt length on multi-domain datasets.** Each domain consists of two clients, and the table presents both the average accuracy within each domain and the overall global accuracy.

| Methods | Office31 | | | | OfficeHome | | | | |
|---|---|---|---|---|---|---|---|---|---|
| | A | D | W | Avg | A | C | P | R | Avg |
| Zero-Shot CLIP (Radford et al., 2021) | 80.96 | 71.63 | 74.60 | 75.73 | 84.21 | 66.37 | 89.16 | 89.68 | 82.35 |
| PromptFL (Guo et al., 2023b) | 88.20 | 84.89 | 91.14 | 88.08 | 86.75 | 75.30 | 94.38 | 93.24 | 87.41 |
| Prompt+Prox (Li et al., 2020) | 88.32 | 85.03 | 91.43 | 88.26 | 86.58 | 75.65 | 94.79 | 93.27 | 87.57 |
| FedOTP (Li et al., 2024a) | 86.62 | 87.40 | 93.55 | 89.19 | 80.38 | 76.29 | 92.49 | 87.86 | 84.26 |
| FedPGP (Cui et al., 2024) | 89.55 | 90.70 | 94.50 | 91.58 | 88.34 | 78.09 | 95.49 | **93.86** | 88.95 |
| FedPHA | **90.44** | **95.80** | **97.98** | **94.74** | **88.70** | **79.59** | **95.93** | 93.83 | **89.51** |

strains local prompt updates using a proximal term instead of direct aggregation. Additionally, we include two popular methods that integrate both global and local prompts: (4) FedOTP (Li et al., 2024a), which employs the Unpaired Optimal Transport (UOT) method to align prompts with the most relevant image features, thereby enhancing personalization. (5) FedPGP (Cui et al., 2024), which uses low-rank decomposition and contrastive learning to balance personalization and generalization.

**Implementation Details.** All methods use a frozen CLIP model with two backbones: ResNet50 (He et al., 2016) and ViT-B16 (Dosovitskiy et al., 2021), with ViT-B16 as the default. Local training rounds are set to $E = 1$ and federated communication rounds to $R = 50$, except for CIFAR-10 and CIFAR-100, where $R = 25$. Final performance is averaged over the last 10 communication rounds. The number of clients varies by dataset. CLIP datasets (Food101, DTD, Caltech101, Flowers102, OxfordPets) use $N = 10$, with each client holding a distinct class subset. Multi-domain datasets (Office31, OfficeHome) set $N$ to twice the number of domains, assigning each domain's data to two clients. CIFAR-10 and CIFAR-100 use $N = 100$, with each client randomly assigned 10% of the dataset. For learnable prompts, the default length is 16 with a 512-dimensional representation. In heterogeneous settings, local prompt lengths range from 4 to 32, while the global prompt length remains 16. The batch size is 32 for training and 128 for testing. For hyperparameter settings, the ratio ($\rho$ in

Eq.(9)) defaults to 0.8, and alpha ($\alpha$ in Eq.(12)) to 1. More details are provided in Appendix Section B.2.

### 4.2. Comparison with State-of-the-Art Methods

**Evaluation Protocol.** We evaluate the models on each client's private test data, which follows the same distribution as its training set. The reported results represent the average test accuracy across all clients over three different seeds. For fairness, we use the same prompt length as other models for comparison.

**Single-Domain Model Evaluation.** To verify that the proposed method remains effective with a fixed prompt length, we first evaluate FedPHA against baseline methods on single-domain CLIP datasets under a pathological non-IID setting. For ease of comparison, Table 1 presents results using the 16-shot setting. As shown in the table, FedPHA consistently outperforms state-of-the-art algorithms across all datasets, demonstrating the effectiveness of our global-local prompt separation mechanism in single-domain scenarios. Notably, on the Food101 dataset, FedPHA achieves a 3.46% performance gain over the best competing method, further highlighting its superiority. An analysis of convergence speed is provided in Appendix Section C.1.

**Impact of Number of Shots.** Additionally, we explore the impact of the number of shots on FedPHA. To analyze this, we vary the number of shots during training from $[1, 2, 4, 8, 16]$. As shown in Figure 3, FedPHA consistently

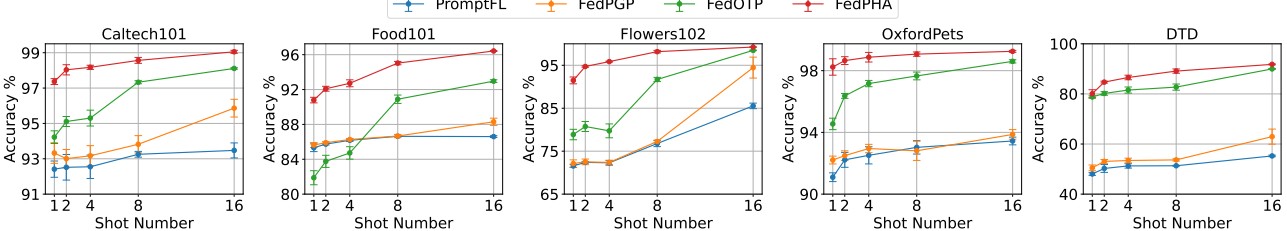

Figure 3: **Ablation study on the number of shots.** The x-axis represents the number of shots, and the y-axis denotes the average test accuracy. Each curve corresponds to a different method, with error bars indicating standard deviations across multiple random seeds.

Table 3: **Runtime overhead of SVD-based prompt projection.** All results are averaged over 10 communication rounds. Compared to model training, the SVD overhead is negligible

| Operation Stage | Time (ms) | Overhead |
|---|---|---|
| Global prompt decomposition | 4.2 | <**1%** |
| Local prompt projection | 2.8 | <**1%** |
| Local model training | 4536.8 | — |

outperforms other methods across all shot settings. In particular, when the number of shots is small, other methods experience significant performance degradation, whereas FedPHA exhibits only a slight decline compared to its performance at 16 shots. This demonstrates the robustness of our approach, enabling effective and rapid adaptation to personalized client requirements even in few-shot scenarios.

**Multi-Domain Model Evaluation.** We also evaluate the performance of FedPHA in comparison to baseline methods on multi-domain datasets. To simulate client heterogeneity, we partition data within the same domain into two clients using a Dirichlet distribution ($\beta = 0.5$). We analyze both the average performance of clients within the same domain and the overall mean performance across all clients. Results for the Office31 and OfficeHome datasets are summarized in Table 2. Our method consistently outperforms baseline approaches. For instance, on the Office31 dataset, FedPHA outperforms FedPGP across all domains, further validating its effectiveness in handling heterogeneous data distributions. These results demonstrate the robustness of FedPHA under diverse domain settings.

**Computational Cost of SVD.** To evaluate the efficiency of the proposed SVD-based prompt projection mechanism, we measure its runtime cost on the client side. As summarized in Table 3, the additional overhead introduced by SVD is minimal compared to standard model training. Specifically, global prompt decomposition and local prompt projection take 4.2 ms and 2.8 ms per communication round on average, each contributing less than 1% to the total training time. Importantly, these operations are performed only once per round, not per batch, making their amortized cost negligible. In contrast, local model training—including forward

and backward passes—dominates the runtime, taking over 4 seconds per round. These results indicate that the added SVD step does not introduce any significant computational bottleneck, even under large-scale settings. This lightweight design confirms that the benefits of SVD-based global-local prompt separation come at almost no cost, further supporting the practicality of our method in federated settings where efficiency is critical.

### 4.3. Effectiveness of prompt length heterogeneity

**Evaluation Protocol.** In this set of experiments, each client uses a different prompt length ranging from 4 to 32. For multi-domain datasets, the specified prompt lengths are applied, whereas for CIFAR-10/100, each client is assigned a randomly selected prompt length.

**Cross Domain Analysis.** We investigate the impact of different prompt length combinations on cross-domain performance. In the Office31 dataset with three domains, each domain has eight possible prompt length choices: [4, 8, 12, 16, 20, 24, 28, 32]. After evaluating 512 combinations, we identify the optimal combination as [28, 12, 16], achieving an accuracy of 95.45%. Figure 4 visualizes the results, where the color intensity of each cell represents the global accuracy for a given prompt length combination. Black-outlined cells indicate cases where all domains use the same prompt length. Notably, most high-accuracy points fall outside these black-boxed regions, suggesting that the conventional approach of assigning the same prompt length to all clients is suboptimal and fails to capture the varying prompt length requirements introduced by data heterogeneity. In contrast, FedPHA enables each client to adopt different prompt lengths, demonstrating its effectiveness in handling heterogeneous data distributions. More details and additional experiments on the OfficeHome dataset are provided in Appendix Section C.2.

**Intra Domain Analysis.** Furthermore, we explore the impact of prompt length on client performance across different length combinations. The two clients within the same domain use identical prompt lengths. As shown in Figure 5, we illustrate the effect of prompt lengths on the performance

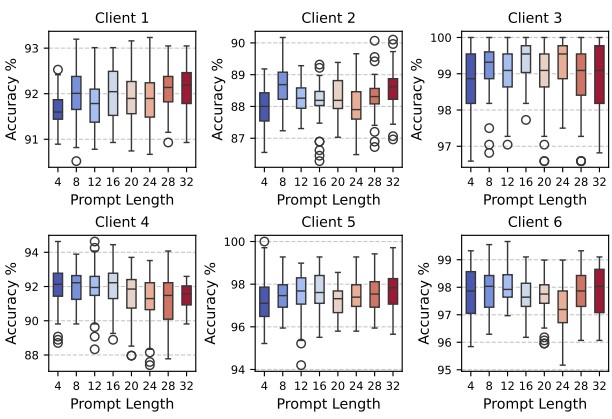

Figure 4: **Impact of Prompt Length on Domains Performance** across different length combinations on overall accuracy. Each 8×8 heatmap illustrates the effect of different Domain 2 and Domain 3 prompt length combinations on model performance when Domain 1 has a fixed prompt length in Office31 dataset. The X-axis represents the average prompt length of Domain 2, while the Y-axis represents that of Domain 3. Color intensity indicates accuracy, with red representing higher accuracy and blue representing lower accuracy. Black-boxed grids highlight cases where the prompt length of Domain 1 matches that of Domain 2 or 3 under the current length combination.

Figure 5: **Impact of Prompt Length on Client Performance** across different length combinations. Clients (1,2), (3,4), and (5,6) belong to the same domain, respectively. The X-axis represents the prompt length, while the Y-axis shows the client's accuracy. The box depicts the IQR, capturing the middle 50% of values, with the horizontal line inside indicating the median accuracy. Whiskers extend to the min/max within 1.5 times the IQR, and outliers appear as points beyond them, highlighting deviations.

Table 4: **Comparison with the SOTA methods and FedPHA variants (fixed vs. random prompt length) on CIFAR-10 and CIFAR-100** across 100 clients. All baseline methods use a fixed prompt length of 16. FedPHA (fixed length) also uses 16 tokens for all clients, while FedPHA (random length) assigns each client a random prompt length between 4 and 32.

| Methods | CIFAR-10 | CIFAR-100 |
|---|---|---|
| CLIP (Radford et al., 2021) | 87.88±0.11 | 64.89±0.19 |
| PromptFL (Guo et al., 2023b) | 91.70±0.11 | 72.58±0.04 |
| Prompt+Prox (Li et al., 2020) | 91.83±0.12 | 72.08±0.09 |
| FedPGP (Cui et al., 2024) | 92.10±0.21 | 74.81±0.48 |
| FedOTP (Li et al., 2024a) | 93.43±0.41 | 75.07±0.39 |
| FedPHA (fixed length) | **94.11±0.14** | **75.92±0.13** |
| FedPHA (random length) | 93.80±0.17 | 75.63±0.17 |

of each client. We observe that clients within the same domain exhibit similar sensitivity to prompt length, even when their data distributions differ. For instance, both Client 1 and Client 2 achieve the best performance with a prompt length of 32, while the worst performance occurs at a prompt length of 4. However, the sensitivity to prompt length varies across domains. For example, while Client 1's optimal prompt length is 32, Client 3 performs best at a prompt length of 16 and performs poorly at 32.

**Robustness Analysis.** On the CIFAR datasets, we randomly select prompt lengths to evaluate robustness. Table 4 compares state-of-the-art methods under the Dirichlet setting ($\beta = 0.5$). FedPHA achieves the best performance on both datasets, demonstrating strong generalization in non-IID scenarios. Its improvement on CIFAR-100 further highlights the effectiveness of the global-local prompt separation mechanism in balancing personalization and global performance. To examine prompt length adaptability, we compare FedPHA with fixed-length (16 tokens) and random-length

(4–32 tokens) prompts. The fixed-length variant slightly outperforms the random one, not due to design limitations, but because random lengths may not align with each client's data distribution, leading to occasional inefficiencies. In contrast, fixed lengths ensure stable optimization and aggregation. Nevertheless, the random-length setting better reflects real-world client heterogeneity, and FedPHA is uniquely capable of operating under such conditions. Future work may explore adaptive prompt assignment strategies to further improve performance in heterogeneous environments.

## 5. Conclusion

This paper proposes a novel and effective method of FedPHA for federated prompt learning. FedPHA is capable of handling heterogeneity problem and alleviating conflicts between global and local knowledge. In particular, we design a G-L heterogeneous federated prompt architecture to effectively accommodate varying prompt lengths. Meanwhile, we introduce SVD-based projection and bidirectional alignment to reduce the negative impact caused by the conflict between global and local prompts. Experimental results on classification tasks demonstrate that our method outperforms state-of-the-art approaches and validate the effectiveness of prompt length heterogeneity.

## Acknowledgement

This work is supported by the National Key Research and Development Program of China (2023YFC2705700), and National Natural Science Foundation of China under Grant (62361166629, 62176188, 62225113, 623B2080), the Wuhan University Undergraduate Innovation Research Fund Project. The supercomputing system at the Supercomputing Center of Wuhan University supported the numerical calculations in this paper.

## Impact Statement

This paper presents work whose goal is to advance the field of Machine Learning. There are many potential societal consequences of our work, none which we feel must be specifically highlighted here.

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

# A. Method Details

## A.1. Notations Definition

To facilitate understanding of our proposed FedPHA, we provide a summary of key notations used throughout this paper in Table 5.

Table 5: **Summary of Notations.**

| Symbol | Definition | Symbol | Definition |
|--------|------------|--------|------------|
| $N$ | Clients number | $\mathcal{L}$ | Loss function |
| $i$ | Client index | $\alpha$ | Push loss margin |
| $\mathcal{D}_i$ | Local dataset | $U, S, V$ | SVD matrices |
| $n_i$ | Dataset size | $\rho$ | Null-space ratio |
| $\mathcal{C}_r$ | Clients in round $r$ | $d'$ | Null-space dim. |
| $E$ | Local epochs | $V_2$ | Null-space basis |
| $P_{g,i}^{r,e}$ | Global prompt | $Q$ | Projection matrix |
| $P_{l,i}^{r,e}$ | Local prompt | $\tilde{Y}_p$ | CLIP text input |
| $\widetilde{P}_{l,i}^{r,e}$ | Projected prompt | $t_{\text{Padding}}$ | Zero padding |
| $P_g^{(r+1,0)}$ | Aggregated prompt | $g(\cdot)$ | Text encoder |
| $T_g$ | Global prompt length | $f(\cdot)$ | Image encoder |
| $T_i$ | Local prompt length | $\tilde{g}_p$ | Text feature |
| $\eta$ | Learning rate | $\tilde{f}$ | Image feature |
| $\tau$ | Softmax temperature | $\text{sim}(\cdot, \cdot)$ | Cosine similarity |

## A.2. Discussion

**Distinction from Existing G-L Prompt Methods.** Here, we provide a more detailed analysis of how our approach fundamentally differs from prior work in terms of architecture design and personalization flexibility.

- **FedOTP** (Li et al., 2024a) adopts a dual-prompt structure consisting of a global prompt and a local prompt. However, it requires both prompts to be of equal length due to the constraints of its unbalanced optimal transport framework. This architectural constraint significantly limits flexibility and the ability to tailor local representations to client-specific needs.

- **FedPGP** (Cui et al., 2024) employs a global prompt alongside two local adapters. In this setup, the local prompt is generated by adding a local adapter to the global prompt, resulting in a tightly coupled formulation. This additive dependency forces the local prompt to inherit features from the global prompt, which may be suboptimal. In scenarios where the global prompt is poorly aligned with a client's local data, this coupling can lead to negative transfer, reducing the effectiveness of personalization.

- **FedPHA (Ours)** introduces a decoupled G-L prompt structure where each client receives a shared fixed-length global prompt and independently configures its local prompt with a variable length. This design explicitly supports heterogeneous prompt configurations, enabling better alignment with diverse data distributions and computational capacities. Importantly, by decoupling global

and local prompts, FedPHA avoids negative transfer from global representations to client-specific learning, thereby enhancing the robustness and adaptability of personalization in federated settings.

# B. Experimental Details

## B.1. Details of Dataset Setup

For our evaluation, we selected nine diverse visual classification datasets as benchmarks. Table 6 provides a detailed overview, including the original task, number of classes, training and test sample sizes, and domain counts.

For datasets with multiple domains, we followed the well-established Office-31 benchmarking protocol, which includes three domains: Amazon (A), Webcam (W), and DSLR (D). These domains represent variations in image quality and style, capturing differences between online product images (Amazon), low-resolution webcam photos, and high-resolution DSLR images. Additionally, we incorporated Office-Home, which consists of four domains: Art (Ar), Clipart (Cl), Product (Pr), and Real-World (Rw). These domains encompass diverse image sources, including artistic renderings, clipart illustrations, product photos, and natural scene images, effectively capturing distribution shifts across different acquisition methods and environments.

By leveraging these datasets, our evaluation ensures a comprehensive assessment of model performance across varying domains and real-world conditions.

## B.2. Details of Implementation

The optimizer used is Stochastic Gradient Descent (SGD) (Robbins & Monro, 1951) with a learning rate of $\eta = 0.001$. All input images are resized to $224 \times 224$ pixels and further divided into $14 \times 14$ patches with a dimension of 768. We conducted all experiments with PyTorch (Paszke et al., 2019) on NVIDIA RTX 3090 GPUs.

## B.3. Details of Baseline Implementation

To ensure fair and transparent comparison with existing Global-Local prompt methods, we re-implemented both FedOTP (Li et al., 2024a) and FedPGP (Cui et al., 2024) under a unified experimental framework. All experiments followed the protocol described in Section 4.1 and Appendix B.2, including identical training schedules, model architectures, and optimization settings.

For all methods, we used a frozen CLIP backbone (ViT-B/16), with a prompt length of 16 and embedding dimension 512. Local training was performed using SGD with a learning rate of 0.001 and a batch size of 32. Each client trained for one local epoch per round, and we ran 50 communication rounds (reduced to 25 for CIFAR-10 and CIFAR-100). In ad-

Table 6: **Statistical details of datasets used in experiments.**

| Dataset | Classes | Train | Test | Domains | Task |
|---|---|---|---|---|---|
| Caltech101 (Fei-Fei, 2004) | 100 | 4,128 | 2,465 | 1 | Object recognition |
| Food101 (Bossard et al., 2014) | 101 | 50,500 | 30,300 | 1 | Fine-grained food recognition |
| Flowers102 (Nilsback & Zisserman, 2008) | 102 | 4,093 | 2,463 | 1 | Fine-grained flower recognition |
| OxfordPets (Parkhi et al., 2012) | 37 | 2,944 | 3,669 | 1 | Fine-grained pet recognition |
| DTD (Cimpoi et al., 2014) | 47 | 2,820 | 1,692 | 1 | Texture classification |
| Office31 (Saenko et al., 2010) | 31 | 3,292 | 813 | 3 | Multi-domain image recognition |
| OfficeHome (Venkateswara et al., 2017) | 65 | 12,475 | 3,113 | 4 | Multi-domain image recognition |
| CIFAR10 (Krizhevsky et al., 2010) | 10 | 50,000 | 10,000 | 1 | General image classification |
| CIFAR100 (Krizhevsky & Hinton, 2009) | 100 | 50,000 | 10,000 | 1 | General image classification |

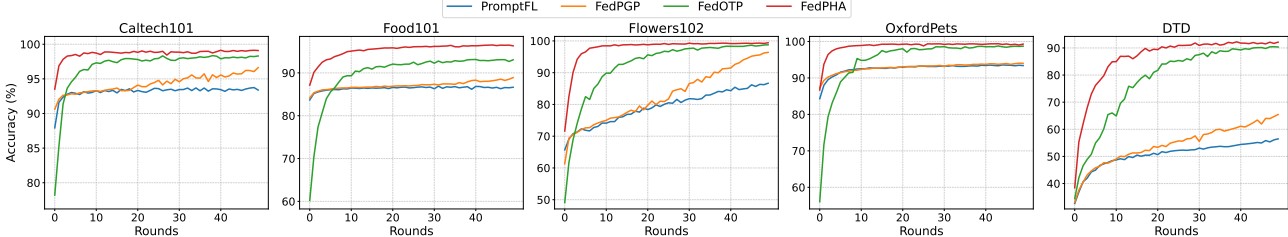

Figure 6: **Comparison with the SOTA methods of convergence speed** on single-domain datasets across 10 clients. The x-axis represents training rounds (from 0 to 50), while the y-axis shows the model accuracy over the course of training.

dition to the shared setup, each baseline has its own method-specific parameters. For FedOTP, we used an unbalanced optimal transport formulation (COT), with Sinkhorn parameters set to `THRESH = 1e-3` and `EPS = 0.1`, and a maximum iteration limit of 100. For FedPGP, we followed its original implementation using a bottleneck dimension of 4 and contrastive loss parameters $\mu = 1$ and temperature $= 0.5$. Other hyperparameters were kept consistent across both baselines, including disabling context initialization (`CTX_INIT = False`), disabling class-specific context prompts (`CSC = False`), using mixed-precision training (`fp16`), and placing the class token at the end of the sequence.

These configurations ensure that any observed performance differences arise from algorithmic or architectural factors, rather than inconsistencies in training conditions. All experiments were repeated across three random seeds for statistical robustness.

## C. Additional Experiments Results

### C.1. Convergence Analysis

Figure 6 presents a comparison of convergence speed across five different datasets (Caltech101, Food101, Oxford Flowers, Oxford Pets, and DTD) over 50 training rounds. Each subfigure represents a distinct dataset, illustrating the per-

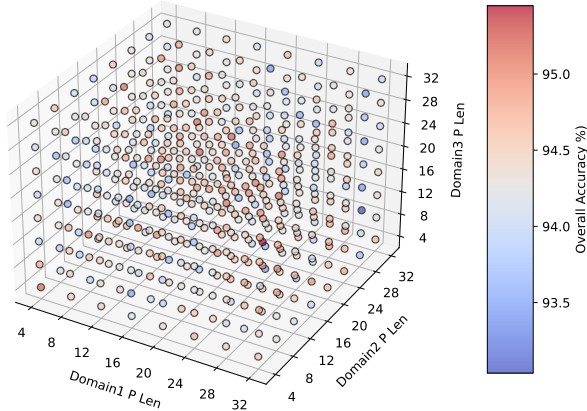

Figure 7: **Prompt length combination effects on Office31.** The X, Y, and Z axes represent the average prompt lengths for Domain 1, Domain 2, and Domain 3, respectively. The color intensity of each point indicates the global accuracy achieved under the corresponding prompt length configuration.

formance of four different training methods: PromptFL, FedPGP, FedOTP, and FedPHA. It can be observed that our FedPHA consistently converges faster than other methods, and its accuracy remains higher than that of other approaches at all stages. This demonstrates the effectiveness of our method in personalization.

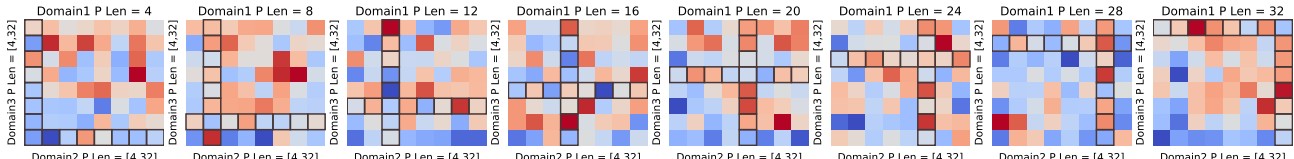

Figure 8: **Impact of Prompt Length on Domain Performance in OfficeHome.** Each 8×8 heatmap illustrates the effect of different Domain 2 and Domain 3 prompt length combinations on global accuracy, when the prompt length of Domain 1 is fixed and Domain 4 is set to 16. The X-axis represents the prompt length of Domain 2, while the Y-axis represents that of Domain 3. Color intensity indicates accuracy, with red representing higher accuracy and blue representing lower accuracy. Black-boxed grids highlight configurations where the prompt length of Domain 1 matches that of Domain 2 or 3.

## C.2. Inter Domain Analysis

Table 7: **Comparison of Overall Mean values across different domains with different prompts lengths.**

| ID | Domain1 | Domain2 | Domain3 | Overall Acc |
|----|---------|---------|---------|-------------|
| 1  | 28 | 12 | 16 | 95.45 |
| 2  | 20 | 12 | 28 | 95.19 |
| 3  | 24 | 12 | 32 | 95.15 |
| 4  | 4  | 4  | 4  | 95.13 |
| 5  | 8  | 32 | 16 | 95.12 |
| 6  | 32 | 8  | 28 | 95.12 |
| 7  | 28 | 4  | 28 | 95.09 |
| 8  | 16 | 32 | 8  | 95.05 |
| 9  | 8  | 16 | 32 | 95.05 |
| 10 | 20 | 4  | 32 | 95.04 |
| ⋮  | ⋮  | ⋮  | ⋮  | ⋮     |
| 512 | 32 | 28 | 12 | 93.04 |
| Mean of 512 combinations | | | | 94.36 |

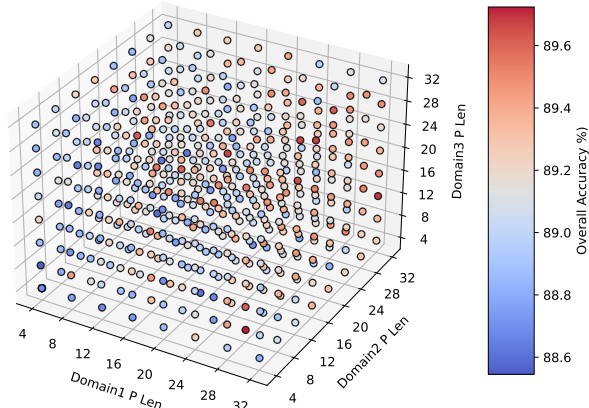

Figure 9: **Prompt length combination effects on OfficeHome.** The X, Y, and Z axes represent the average prompt lengths for Domain 1, Domain 2, and Domain 3, respectively, with Domain 4 fixed at length 16. The color intensity of each point reflects the global accuracy.

Additionally, we investigated the impact of different prompt length combinations on overall performance, as shown in Figure 7. The eight heatmaps in Figure 4 can be regarded as planar slices of Figure 7, providing a more granular view of how prompt length variations influence accuracy across different domains. In Table 7, we list the average accuracy of 512 prompt length combinations sorted in descending order. From the table, we can observe that the only well-performing combination with uniform lengths is [4,4,4], while all other combinations have varying lengths. This further validates our research motivation: for clients with different data distributions, adopting different adaptive prompt lengths is more beneficial for personalization.

Beyond Office31, we extended our inter-domain prompt length analysis to the OfficeHome dataset, which comprises four distinct domains. Given the increased number of clients, the total number of possible prompt length combinations grows exponentially ($8^4 = 4096$), making exhaustive evaluation computationally infeasible. To reduce complexity while maintaining representative coverage, we constrained the prompt length of the fourth domain to 16, thereby nar-

rowing the search space to a manageable 512 combinations.

The corresponding 3D scatter plot is illustrated in Figure 9, where each point represents a unique prompt length combination for the first three domains. The spatial distribution reveals clear patterns: high-performing configurations tend to cluster in regions where the prompt lengths differ across domains, again suggesting that uniform configurations are not optimal. This observation is consistent with our earlier findings from Office31.

In addition, Figure 8 presents eight heatmaps with fixed prompt lengths for Domain1, offering a slice-by-slice visualization across Domain2 and Domain3. Similar to Office31, black-boxed cells indicate uniform prompt lengths across all domains. These boxed areas rarely correspond to the highest accuracy regions, reinforcing the insight that heterogeneity-aware prompt length selection is critical for maximizing performance in multi-domain federated settings.

Taken together, these findings provide converging evidence across datasets: the assumption that all clients benefit from the same prompt length is fundamentally flawed. Instead,

Table 8: **Sensitivity analysis of projection ratio $\rho$ and push margin $\alpha$ on 5 datasets.** FedPHA achieves robust performance across a wide range of settings with shot number = 16. Default values are $\rho = 0.8$, $\alpha = 1.0$.

(a) Average over 5 datasets.

| $\alpha$ / $\rho$ | 0.3 | 0.5 | 0.8 | 1.0 | Avg. |
|---|---|---|---|---|---|
| 0.5 | 97.05 | 97.04 | 96.99 | 96.92 | 97.00 |
| 1.0 | 97.04 | 97.02 | **97.13** | 97.08 | 97.07 |
| 1.5 | 96.94 | 96.99 | 96.94 | 96.95 | 96.96 |
| 2.0 | 96.88 | 96.82 | 96.85 | 96.87 | 96.85 |
| Avg. | 96.98 | 96.97 | 96.98 | 96.96 | 96.97 |

(b) Caltech101.

| $\alpha$ / $\rho$ | 0.3 | 0.5 | 0.8 | 1.0 | Avg. |
|---|---|---|---|---|---|
| 0.5 | 98.97 | 98.99 | 98.91 | 98.97 | 98.96 |
| 1.0 | 99.01 | 99.05 | **99.12** | 99.04 | 99.06 |
| 1.5 | 98.97 | 99.07 | 98.93 | 98.95 | 98.98 |
| 2.0 | 98.89 | 98.77 | 98.88 | 98.84 | 98.84 |
| Avg. | 98.96 | 98.97 | 98.96 | 98.95 | 98.96 |

(c) Food101.

| $\alpha$ / $\rho$ | 0.3 | 0.5 | 0.8 | 1.0 | Avg. |
|---|---|---|---|---|---|
| 0.5 | 96.22 | 96.27 | 96.23 | 96.27 | 96.25 |
| 1.0 | 96.35 | 96.38 | 96.42 | 96.33 | 96.37 |
| 1.5 | **96.61** | 96.51 | 96.49 | 96.48 | 96.52 |
| 2.0 | 96.48 | 96.42 | 96.43 | 96.38 | 96.43 |
| Avg. | 96.42 | 96.39 | 96.39 | 96.36 | 96.39 |

(d) Flowers102.

| $\alpha$ / $\rho$ | 0.3 | 0.5 | 0.8 | 1.0 | Avg. |
|---|---|---|---|---|---|
| 0.5 | 99.33 | 99.16 | 99.23 | 99.24 | 99.24 |
| 1.0 | 99.28 | 99.19 | 99.23 | **99.32** | 99.25 |
| 1.5 | 99.01 | 99.14 | 99.14 | 99.12 | 99.10 |
| 2.0 | 99.17 | 98.78 | 99.08 | 99.07 | 99.02 |
| Avg. | 99.20 | 99.07 | 99.17 | 99.19 | 99.16 |

(e) OxfordPets.

| $\alpha$ / $\rho$ | 0.3 | 0.5 | 0.8 | 1.0 | Avg. |
|---|---|---|---|---|---|
| 0.5 | 99.15 | 99.06 | 99.13 | 99.08 | 99.11 |
| 1.0 | 99.24 | 99.18 | 99.21 | 99.14 | 99.19 |
| 1.5 | 99.08 | **99.27** | 99.03 | 99.06 | 99.11 |
| 2.0 | 99.06 | 98.94 | 98.86 | 99.01 | 98.97 |
| Avg. | 99.13 | 99.11 | 99.06 | 99.07 | 99.09 |

(f) DTD.

| $\alpha$ / $\rho$ | 0.3 | 0.5 | 0.8 | 1.0 | Avg. |
|---|---|---|---|---|---|
| 0.5 | 91.59 | 91.74 | 91.45 | 91.02 | 91.45 |
| 1.0 | 91.31 | 91.30 | **91.67** | 91.57 | 91.46 |
| 1.5 | 91.02 | 90.96 | 91.12 | 91.16 | 91.06 |
| 2.0 | 90.82 | 91.17 | 90.98 | 91.07 | 91.01 |
| Avg. | 91.19 | 91.29 | 91.31 | 91.20 | 91.25 |

our FedPHA approach, which allows client-specific prompt length adaptation, effectively captures domain-level variability and leads to superior cross-domain generalization.

### C.3. Sensitivity Analysis

Table 8 reports a detailed sensitivity analysis of FedPHA with respect to the projection ratio $\rho$ in Eq.(9) and the push margin $\alpha$ in Eq.(12) across five datasets. Each cell in the table presents the average accuracy over the last 10 training epochs under a specific configuration.

As shown in Table 8 (a), the average accuracy across all datasets remains consistently high across a broad range of $(\alpha, \rho)$ combinations. The best overall performance (97.13%) is achieved when $\alpha = 1.0$ and $\rho = 0.8$, which are also the default values used throughout our main experiments. Importantly, performance degradation remains minimal—typically within 0.2%—even when deviating from this configuration. This reflects the robustness of FedPHA and its insensitivity to moderate variations in hyperparameter settings, which is a desirable property in practical federated

deployments where fine-tuning may not always be feasible.

For Caltech101 and OxfordPets, performance remains remarkably stable, exceeding 98.8% across all tested settings, with optimal configurations coinciding with the default values. Food101 exhibits a mild preference for larger $\alpha$ (especially 1.5), indicating that a stronger push margin may help in fine-grained classification tasks. Flowers102 shows high tolerance to both $\alpha$ and $\rho$ changes, suggesting that the method generalizes well in image domains with intra-class similarity. In contrast, DTD reveals greater sensitivity to hyperparameter shifts. Nevertheless, its accuracy remains within a relatively narrow and acceptable range (90.8%–91.7%), demonstrating that FedPHA retains competitiveness even under more challenging visual domains.

This table-based analysis confirms that FedPHA achieves strong and stable performance under a wide spectrum of hyperparameter choices. The default configuration ($\alpha$=1.0, $\rho$=0.8) serves as a robust and well-balanced setting across diverse data distributions, minimizing the need for extensive tuning in real-world federated learning scenarios.

