# OpenReview forum: "FedPHA: Federated Prompt Learning for Heterogeneous Client Adaptation"
_ICML.cc/2025/Conference — ICML 2025 poster_

### Official Review · Reviewer_TNUk · 2025-02-14

**Overall Recommendation:** 4

**Summary:**

This paper proposes a method called FedPHA (Federated Prompt Learning for Heterogeneous Client Adaptation) to enhance federated prompt learning in diverse client environments. It addresses two key challenges: the limitation of uniform prompt lengths in existing methods and the conflict between global and local knowledge during aggregation. FedPHA introduces a federated prompt heterogeneous architecture, combining a fixed-length global prompt for efficient aggregation with variable-length local prompts to capture client-specific characteristics. To further mitigate conflicts, FedPHA incorporates SVD-based projection to filter out conflicting information and bidirectional alignment to maintain local distinctiveness while benefiting from global knowledge. The approach is communication-efficient, requiring no major modifications to pre-trained models, and significantly improves performance in federated learning settings.

## update after rebuttal
The paper is well-motivated and clearly structured, presenting a novel and effective solution to the practical Federated Learning problem. After carefully reading the rebuttal, my concerns have been sufficiently addressed. Therefore, I maintain my positive rating.

**Claims And Evidence:**

Yes, the claims in the submission are supported by experimental results.

**Essential References Not Discussed:**

No essential references appear to be missing.

**Experimental Designs Or Analyses:**

The experimental design is well-structured and valid, evaluating the proposed methods on diverse heterogeneous datasets with non-IID settings. Comparisons with five baseline methods and clearly defined hyperparameters ensure a robust assessment.

**Methods And Evaluation Criteria:**

The methods and evaluation criteria are appropriate for addressing client heterogeneity.

**Other Comments Or Suggestions:**

The method proposed in this paper enables federated prompt learning to adapt to varying local prompt lengths. I hope future work can explore this aspect further.

**Other Strengths And Weaknesses:**

Strengths:
1. This approach tackles client heterogeneity in federated learning, improving adaptability to diverse data distributions. FedPHA is communication-efficient and scalable, requiring no extensive fine-tuning for real-world applications.
2. The motivation is clear and compelling. The federated prompt heterogeneous architecture enables aggregation while accommodating individual client needs. The SVD-based projection mechanism resolves conflicts between global and local knowledge, preserving essential local information. The bidirectional alignment function ensures alignment and preserves the unique characteristics of both global and local representations.
3. The experiment is comprehensive, covering a wide range of scenarios to thoroughly evaluate the proposed methods.

Weaknesses:
1. In Section 3.2, according to the submitted code, zero padding is applied at the end of the sequence, while the paper incorrectly describes it as being added to the middle. This inconsistency between the implementation and the description should be corrected.
2. In Section 4.3, the experiments are only conducted on the Office31 dataset. It would be beneficial to include more datasets in this section to provide a more comprehensive evaluation.
3. In Table 3, a comparison with a FedPHA model using a fixed local prompt length should be added. This comparison would help demonstrate the robustness of using random prompt lengths in FedPHA.

**Questions For Authors:**

This paper proposes an innovative FPL approach. While the methodology is comprehensive, there are several concerns regarding the implementation details, additional dataset evaluation, and prompt lengths. Please address these concerns in the rebuttal.

**Relation To Broader Scientific Literature:**

Existing federated prompt learning methods cannot support different local prompt lengths due to aggregation requirements and structural constraints. In contrast, FedPHA introduces a federated prompt heterogeneous architecture, combining a fixed-length global prompt for efficient aggregation with variable-length local prompts to capture client-specific characteristics.

**Theoretical Claims:**

In this paper, the SVD-based projection mechanism has a solid theoretical foundation, and no significant issues were identified in the proofs.

---

> ### Author Rebuttal · Authors · 2025-03-31
>
> Dear Reviewer TNUk:
>
> We sincerely thank the reviewer for the constructive and encouraging feedback. We are especially grateful for your positive recognition of our contributions to federated prompt learning, the novelty of the proposed architecture, and the overall experimental design. Below, we address the specific concerns raised in the Weaknesses section:
>
> ### Weakness
>
> **W1: Correction of Padding Description**
>
> A1: Thank you for pointing out this discrepancy. You are correct that the current implementation applies **zero-padding at the end** of the prompt sequence, while the paper previously stated it was added to the middle. We apologize for this inconsistency. We have now **Corrected the description** in Section 3.2 to accurately reflect the implementation: zero-padding is appended to the end of the prompt sequence when needed.
>
> **W2: Broadening the Evaluation Scope with Additional Datasets**
>
> A2: Thank you for the thoughtful suggestion. We agree that evaluating the effectiveness of prompt length heterogeneity on additional datasets strengthens the generality of our findings.
>
> To this end, we have extended our analysis in Section 4.3 to include the **OfficeHome** dataset, which comprises four domains: *Art, Clipart, Product,* and *Real-World*. For this experiment, we allowed each domain to select its own prompt length from eight possible values:[4, 8, 12, 16, 20, 24, 28, 32]. We then evaluated the global classification accuracy under each prompt length configuration.
>
> We evaluated global classification accuracy under various prompt length configurations. As shown in the table below, **heterogeneous prompt length settings consistently outperformed uniform ones**. For example, the best-performing configuration on Office31 was [28, 12, 16], achieving **95.45%** accuracy, while the uniform setting [16, 16, 16] reached only 94.72%. Similarly, for OfficeHome, the best result was obtained with [12, 32, 12, 16] at **89.67%**, outperforming uniform settings such as [16, 16, 16, 16] (89.17%) and [32, 32, 32, 32] (88.41%). These consistent trends across datasets further support our claim that **prompt length heterogeneity is beneficial in federated learning scenarios with non-IID data**.
>
> **Table: Accuracy under different prompt length combinations on multi-domain datasets**
>
> | Prompt Lengths [A, W, D] | Office31 (%) | Prompt Lengths [A, C, P, R] | OfficeHome (%) |
> |-|-|-|-|
> | [4, 4, 4] | 95.13 | [4, 4, 4, 4] | 88.73 |
> | [16, 16, 16] | 94.72 |[16, 16, 16, 16] | 89.17 |
> | [32, 32, 32] | 94.05 |[32, 32, 32, 32] | 88.41 |
> | [28, 12, 16] (best) | **95.45** |[12, 32, 12, 16] (best) | **89.67** |
> | [20, 12, 28] | 95.19 | [12, 12, 28, 16] | 89.61 |
> | [24, 12, 32] | 95.15 | [12, 16, 20, 16] | 89.56 |
>
> Due to rebuttal space constraints, we will provide a more detailed and visual analysis (similar to Figure 4 and Figure 5) in the final version of the paper.
>
> **W3: Comparison with Fixed-Length FedPHA**
>
> A3: We thank the reviewer for this valuable suggestion. To evaluate the robustness of FedPHA under prompt length variation, we conducted a direct comparison between two settings: **FedPHA with a fixed local prompt length** (uniformly set to 16 tokens) and **FedPHA with randomly assigned prompt lengths** (ranging from 4 to 32) across clients.
>
> As shown in the table below, the fixed-length version slightly outperforms the variable-length one on both CIFAR-10 and CIFAR-100. This difference may be attributed to the greater **optimization stability** and **aggregation consistency** achieved when all clients share the same prompt structure. In contrast, randomly assigned prompt lengths may **not always align with the specific needs or data distributions of individual clients**, inevitably resulting in minor performance drops.
>
> However, this setting also reflects a more realistic federated environment, where clients may differ significantly in computational capacity or data complexity. Notably, **FedPHA is the only method capable of operating under heterogeneous prompt configurations**, offering unique flexibility and scalability in such scenarios. In more diverse and non-IID datasets (e.g., Office31, OfficeHome), we observe that **randomized or adaptive prompt lengths lead to improved performance**, highlighting the benefits of prompt flexibility in cross-domain and heterogeneous environments.
>
> **Table: Comparison with the SOTA methods using random and fixed prompt lengths on CIFAR-10 and CIFAR-100 across 100 clients.** All methods except FedPHA (last row) use a fixed prompt length of 16.
>
> | Methods|CIFAR-10|CIFAR-100|
> |-|-|-|
> | CLIP|87.88|64.89|
> | PromptFL|91.70|72.58|
> | Prompt+Prox|91.83|72.08|
> | FedPGP|92.10|74.81|
> | FedOTP|93.43|75.07|
> | **FedPHA (fixed length)**|**94.11**|**75.92**|
> | **FedPHA (random length)**|93.80|75.63|

---

### Official Review · Reviewer_oKVN · 2025-03-03

**Overall Recommendation:** 3

**Summary:**

The authors introduce Federated Prompt Learning for Heterogeneous Client Adaptation (FedPHA), a novel approach to adapting pre-trained Vision-Language Models (VLMs) within federated learning. The primary motivation is to tackle the persistent heterogeneity challenge by integrating a uniform global prompt for efficient aggregation with diverse local prompts for personalization. Additionally, FedPHA aims to mitigate conflicts between global and local knowledge, enabling clients to maintain their unique characteristics while leveraging shared information. To achieve this, Singular Value Decomposition (SVD)-based projection and bidirectional alignment are introduced, enhancing model generalization while preserving local optimization. Experimental results confirm FedPHA’s superiority over state-of-the-art methods, striking a balance between personalization and global knowledge in heterogeneous federated learning settings.

**Claims And Evidence:**

Yes, the claims are well-supported by empirical evidence, including extensive experiments and ablation studies. The results consistently demonstrate FedPHA’s superiority over baselines, validating its effectiveness in handling heterogeneity in federated learning.

**Essential References Not Discussed:**

The paper provides a comprehensive review of related work, citing essential studies in federated learning, prompt learning, and vision-language models. No significant omissions were identified.

**Experimental Designs Or Analyses:**

Yes, I reviewed the experimental setup, including dataset partitioning, evaluation metrics, and ablation studies. The methodology is sound, and the comparisons with baselines are fair.

**Methods And Evaluation Criteria:**

Yes, the chosen datasets cover both single-domain and multi-domain scenarios, effectively simulating heterogeneous federated learning environments. The evaluation metrics and comparison with state-of-the-art methods align well with the problem setting.

**Other Comments Or Suggestions:**

No further comments or suggestions.

**Other Strengths And Weaknesses:**

Strengths
(1) The paper effectively combines federated learning and prompt learning, introducing a dual-layer prompt structure that balances global generalization and local personalization. The integration of Singular Value Decomposition (SVD)-based projection and bidirectional alignment is an innovative approach to mitigating conflicts between global and local knowledge.
(2) Heterogeneous federated learning is a critical challenge, and this work provides a scalable and adaptable solution. The ability to support varying prompt lengths per client enhances personalization, making it applicable to real-world federated settings with non-IID data distributions.
(3) The experiments cover multiple benchmark datasets, including single-domain, multi-domain, and cross-domain settings, demonstrating the robustness of FedPHA. Comparisons with state-of-the-art methods further validate its effectiveness.
(4) The paper is well-written, with clear problem formulation, methodology descriptions, and result interpretations. Figures and tables are informative, enhancing readability.

Weaknesses
(1) The explanation of the SVD-based projection is unclear. Could the authors further elaborate on why and how it effectively mitigates conflicts between global and local prompts? A more detailed theoretical discussion would be helpful.
(2) The choice of ratio (ρ) in Eq.(9) and alpha (α) in Eq.(12) plays a crucial role in model performance. A deeper sensitivity analysis of these hyperparameters could provide valuable insights into the stability and adaptability of FedPHA across different federated settings.
(3) There are some inconsistent expressions in the paper, such as the dataset names in Table 1 and Figure 3, which are not uniform. The authors may consider carefully reviewing the wording for consistency to improve clarity and readability.

**Questions For Authors:**

I hope the authors can address the concerns I raised in the Weaknesses section.

**Relation To Broader Scientific Literature:**

The paper builds on existing work in federated learning and prompt learning by addressing heterogeneity using Singular Value Decomposition (SVD)-based projection and bidirectional alignment. It extends prior federated prompt learning approaches by enabling varying local prompt lengths, improving adaptability to non-IID data distributions.

**Theoretical Claims:**

The paper primarily focuses on experimental validation and no major theoretical inconsistencies were identified.

---

> ### Author Rebuttal · Authors · 2025-03-30
>
> Dear Reviewer oKVN:
>
> Thank you for your thoughtful review and for raising key concerns regarding our work. We aim to address your concerns in our detailed responses below, hoping to provide clarity and demonstrate the effectiveness of our proposed approach.
>
> ### Weakness
>
> **W1: Theoretical Explanation of SVD-Based Projection**
>
> A1: Thank you for highlighting this point. We agree that a clearer explanation of the SVD-based projection mechanism is essential, especially given its role in handling heterogeneous client data. In **Section 3.3**, we now provide a more detailed account of its purpose, mathematical formulation, and intuitive significance, which we summarize below.
>
> While the dual-layer prompt architecture (global + local) provides personalization flexibility, an **explicit mechanism is required to regulate the interaction between global and local prompts**, particularly in highly heterogeneous settings where simple alignment or orthogonality may fail. Our **SVD-based projection** addresses this by filtering out potentially **conflicting directions** in the global prompt space.
>
> Formally, we apply **Singular Value Decomposition (SVD)** to the current global prompt matrix, decomposing it into orthonormal components. We then use the **trailing singular vectors (i.e., low-variance directions)** to define a null space that captures information **less dominant in the global prompt**, and potentially misaligned with client-specific distributions. The local prompt is projected onto this subspace, effectively **removing dominant global components** that could interfere with personalization.
>
> To mitigate the risk of over-suppression (i.e., losing important local information), we complement this with a **bidirectional alignment strategy** (Section 3.4). Specifically:
>
> - A **pull loss** ensures that the original and projected local prompts remain close in the feature space, preserving essential local semantics.
> - A **push loss** introduces a lower-bound distance between the local and global prompts, ensuring that local prompts remain sufficiently distinct and do not collapse toward overly generic representations.
>
> Together, these mechanisms strike a balance between **global knowledge integration** and **local specialization**, making the projection process both effective and robust in non-IID federated settings. We wish this richer explanation will improve the clarity and motivation of our design.
>
> **W2: Sensitivity Analysis of Hyperparameters ρ and α**
>
> A2: Thank you for raising this important concern. We agree that analyzing the sensitivity of key hyperparameters is crucial to validating the robustness and adaptability of FedPHA across different federated settings. We conducted additional ablation experiments to investigate the impact of the following two hyperparameters:
>
> - ρ (in Eq. (9)): controls the number of low-rank components used for projecting local prompts into the null space of the global prompt.
> - α (in Eq. (12)): defines the margin in the push loss, ensuring local prompts remain sufficiently distinct from the global prompt.
>
> We varied ρ∈ \{0.5, 0.6, 0.7, 0.8, 0.9\} to confirm the Effect of the Ratio in SVD-Based Projection. A larger ρ keeps more dominant directions from the global prompt, while a smaller ρ filters out more components. We observed that:
>
> - Performance is stable when ρ∈[0.7,0.9], suggesting that filtering out only the weaker global components is sufficient.
> - Extremely low ρ may lead to over-filtering and loss of useful shared information.
>
> **Table. Effect of projection ratio ρ on accuracy**
>
> | ρ | Caltech101 (%) | Office31 (%) | CIFAR-100 (%) |
> |-|-|-|-|
> | 0.5 | 97.88 | 93.21 | 74.91 |
> | 0.6 | 98.42 | 93.96 | 75.26 |
> | 0.7 | 98.86 | 94.02 | **76.37** |
> | 0.8 (default) | **99.05** | 94.74 | 75.63 |
> | 0.9 | 98.93 | **94.91** | 75.32 |
>
> We tested α∈ \{0.5, 1.0, 1.5, 2.0\} to confirm the Effect of the Margin Parameter α in the Push Loss, and found:
>
> - Performance is robust when α∈ [1.0, 1.5].
> - Too small a margin (e.g., 0.5) causes local prompts to collapse toward global prompts, reducing personalization.
> - Too large a margin may hinder knowledge sharing, slightly degrading performance.
>
> **Table. Effect of push margin α on accuracy**
> | α | Caltech101 (%) | Office31 (%) | CIFAR-100 (%) |
> |-|-|-|-|
> | 0.5 | 98.11 | 93.07 | 74.02 |
> | 1.0 (default) | **99.05** | 94.74 | **75.63** |
> | 1.5 | 98.79 | **95.16** | 75.21 |
> | 2.0 | 98.32 | 93.85 | 74.85 |
>
> Note: All experiments use ViT-B/16 backbone and the same federated training setup as described in Section 4.1. The default setting in our main experiments is ρ=0.8, α=1.0.
>
> **W3: Consistency in Terminology and Presentation**
>
> A3: Thank you for catching this inconsistency. We have carefully reviewed and revised all dataset names and references throughout the paper to ensure consistency, especially in Table 1, Figure 3, and the surrounding text. These updates improve the overall clarity and readability of the manuscript.

---

### Official Review · Reviewer_tY2b · 2025-03-04

**Overall Recommendation:** 3

**Summary:**

FedPHA is a novel FPL framework designed to address heterogeneous client adaptation in federated learning. Traditional FPL methods enforce uniform prompt lengths, which limits their adaptability to clients with diverse data distributions. To overcome this limitation, FedPHA proposes a dual-layer prompt architecture consisting of a fixed-length global prompt for efficient aggregation and variable-length local prompts to preserve client-specific knowledge. To mitigate conflicts between global and local knowledge, FedPHA incorporates svd-based projection to filter conflicting information and bidirectional alignment to maintain balance between generalization and personalization. Through these innovations, FedPHA provides a scalable, robust, and effective solution for federated learning across diverse, non-IID clients, ensuring both efficient knowledge sharing and local adaptability. ## update after rebuttal

**Claims And Evidence:**

YES

**Essential References Not Discussed:**

None

**Experimental Designs Or Analyses:**

Experimental results are comprehensive, with evaluations conducted across various datasets and comparisons against multiple baseline methods.

**Methods And Evaluation Criteria:**

Yes, the chosen evaluation metrics align with the standard practices in the field.

**Other Comments Or Suggestions:**

No

**Other Strengths And Weaknesses:**

Pros:
P1: FedPHA addresses client heterogeneity in FPL by introducing a dual-layer prompt architecture, combining a fixed-length global prompt for efficient aggregation with variable-length local prompts to adapt to diverse client data, ensuring better personalization while maintaining federated learning efficiency.
P2: This design enhances personalization without sacrificing global knowledge sharing, enabling clients to retain local characteristics while benefiting from shared representations. This structure ensures better adaptation to non-IID data distributions, making FedPHA more effective in heterogeneous federated learning settings.
P3: FedPHA provides a scalable, adaptable, and robust solution for FPL on non-IID datasets, effectively accommodating diverse client distributions and improving overall model performance compared to existing methods.

Cons:
Q1:Many other methods also employ a Global-Local (G-L) prompt architecture for federated learning. Could the authors provide a more detailed explanation of how their approach fundamentally differs from existing G-L prompt methods?
Q2:Could the authors provide more details on how the baselines were configured such as FedOTP and FedPGP, including their hyperparameter settings and any modifications made for fair comparison?
Q3: The paper lacks ablation experiments to analyze the impact of its key hyperparameters. Could the authors provide additional experiments in this regard?

**Questions For Authors:**

Please refer to Strengths And Weaknesses.

**Relation To Broader Scientific Literature:**

Authors introduce a novel approach to address heterogeneous client adaptation in FL and mitigate conflicts between global and local knowledge in FPL.

**Theoretical Claims:**

Authors provide detailed definitions and analysis in Sec 3, including svd-based projection and bidirectional alignment.

---

> ### Author Rebuttal · Authors · 2025-03-30
>
> Dear Reviewer tY2b:
>
> We sincerely thank the reviewer for the thoughtful and detailed evaluation of our work. We are especially grateful for the recognition of our contributions in addressing client heterogeneity through the dual-layer prompt architecture, the introduction of SVD-based projection and bidirectional alignment, and the strong experimental validations. Below, we address each of the reviewer’s key concerns:
>
> ### Weakness
>
> **Q1: Distinction from Existing G-L Prompt Methods**
>
> A1: Thank you for raising this important point. As mentioned in Lines 55–99 of the *Introduction* and Lines 141–153 of the *Related Work* section, we have cited and briefly discussed other existing works. Here, we provide a more detailed explanation of how our approach differs fundamentally from previous Global-Local (G-L) prompt-based methods in terms of architecture design:
>
> - **FedOTP:** Each client uses a global prompt and a local prompt. However, due to the requirement of the **unbalanced optimal transport framework**, FedOTP enforces that **the global and local prompts must be of equal length**, which restricts flexibility and client adaptability.
> - **FedPGP:** Each client uses a global prompt and **two local adapters**. In this design, the local prompt is constructed by **adding a local adapter to the global prompt**, creating a strong dependency between them. This additive formulation forces the local prompt to inherit characteristics from the global prompt. Consequently, if the global prompt captures features that are misaligned with a client’s local data distribution, the local prompt may be **forced to adapt to irrelevant or even conflicting patterns**, reducing the effectiveness of personalization.
> - **FedPHA (Ours):** Each client uses a fixed-length global prompt and a local prompt of varying lengths. Unlike existing G-L prompt methods that often assume a **uniform prompt length across clients**, FedPHA explicitly supports **client-specific variable-length local prompts**, enabling better alignment with heterogeneous computational capabilities and data distributions across clients. **Moreover, by decoupling the global and local prompts, FedPHA effectively mitigates the negative transfer from global knowledge to client-specific representations**, enhancing the robustness of personalization.
>
> **Q2: Baseline Implementation Details and Fair Comparison**
>
> A2: We appreciate the reviewer’s request for greater transparency regarding the baseline configurations. To ensure a fair and controlled comparison, we implemented **FedOTP** and **FedPGP** following the **same experimental protocol** described in **Section 4.1 (Implementation Details)** and **Appendix B.2 (Details of Implementation)**. Below, we provide a summary of both the shared and method-specific settings:
>
> - **Shared Training Setup:** All methods were evaluated using a **frozen CLIP backbone** (default: ViT-B/16), with **local training epochs** set to E = 1 and **communication rounds** to  R = 50 (reduced to R = 25 for CIFAR-10/100). Training was conducted using **SGD** with a learning rate of 0.001 and a batch size of 32. The default **prompt length** was 16, with a 512-dimensional embedding.
> - **FedOTP-Specific Configuration:**
>   - Optimal Transport type: `COT` (Unbalanced Optimal Transport)
>   - Sinkhorn distance parameters: `THRESH = 1e-3`, `EPS = 0.1`
>   - OT maximum iterations: `MAX_ITER = 100`
> - **FedPGP-Specific Configuration:**
>   - Bottleneck dimension: `BOTTLENECK = 4`
>   - Additional loss parameters: `mu = 1`, `temp = 0.5`
>
> All other parameters, such as context initialization (`CTX_INIT = False`), class-specific context (`CSC = False`), prompt precision (`PREC = "fp16"`), and class token position (`"end"`), were kept consistent across both baselines to match our implementation setup. These configurations ensure that the comparisons are fair and focused solely on the architectural and algorithmic differences.
>
> **Q3: Ablation of Key Hyperparameters**
>
> A3: We thank the reviewer for this valuable suggestion. We have already conducted ablation studies on several key hyperparameters and presented the results in the corresponding sections of the paper:
>
> - **Prompt Length:** The ablation on prompt length has been thoroughly analyzed in **Section 4.3 (Effectiveness of Prompt Length Heterogeneity)**, where we demonstrate that supporting variable-length prompts across clients leads to better performance under heterogeneous settings.
> - **Communication Rounds (R):** We provide an analysis of the effect of communication rounds in **Appendix C.1 (Convergence Analysis)**, showing how FedPHA converges efficiently and consistently across datasets.
> - **Hyperparameters ρ and α:** The ablation studies for the ratio ρ in Eq.(9) and the coefficient α in Eq.(12) are discussed in our response to reviewer **oKVN**, where we detail their influence on model performance and stability.

---

### Official Review · Reviewer_A325 · 2025-03-13

**Overall Recommendation:** 3

**Summary:**

This paper introduces FedPHA, a novel Federated Prompt Learning (FPL) approach that enables heterogeneous client adaptation using Vision-Language Models (VLMs). The key contributions include:
A dual-layer architecture combining a fixed-length global prompt for efficient aggregation and variable-length local prompts for personalization. A Singular Value Decomposition (SVD)-based projection mechanism to resolve conflicts between global and local prompts. A bidirectional alignment strategy to optimize the interaction between global knowledge and local adaptation. Extensive empirical validation on various datasets, demonstrating superior performance over state-of-the-art FL and FPL methods. The paper is technically sound, well-motivated, and presents strong experimental results. However, a few areas need further clarification and improvements, particularly regarding real-world feasibility, communication efficiency, and computational overhead.
## update after rebuttal
Thanks to the authors for their thoughtful and comprehensive rebuttal. I am satisfied with the authors' responses and believe they addressed the concerns raised in the initial review. I maintain my positive evaluation of the paper and recommend acceptance.

**Claims And Evidence:**

yes

**Essential References Not Discussed:**

no

**Experimental Designs Or Analyses:**

yes

**Methods And Evaluation Criteria:**

yes

**Other Comments Or Suggestions:**

See Strengths And Weaknesses

**Other Strengths And Weaknesses:**

Strengths:
1. Unlike existing federated prompt learning (FPL) methods that enforce uniform prompt lengths, FedPHA accommodates varying local prompt lengths, making it more adaptable to heterogeneous clients.
2. The SVD-based projection mechanism effectively filters out conflicting components between global and local knowledge, mitigating negative transfer effects.

Weaknesses:
1.	The method requires sending prompts to a central server, which may still introduce communication overhead despite avoiding full model aggregation.
2.	The SVD decomposition of global prompts adds computational overhead at both the server and client side.
3.	Since activations and prompts are exchanged, potential privacy risks (e.g., gradient leakage, prompt inversion attacks) could arise.

**Questions For Authors:**

See Strengths And Weaknesses

**Relation To Broader Scientific Literature:**

This paper introduces FedPHA, a novel Federated Prompt Learning (FPL) approach that enables heterogeneous client adaptation using Vision-Language Models (VLMs). The key contributions include:
A dual-layer architecture combining a fixed-length global prompt for efficient aggregation and variable-length local prompts for personalization. A Singular Value Decomposition (SVD)-based projection mechanism to resolve conflicts between global and local prompts. A bidirectional alignment strategy to optimize the interaction between global knowledge and local adaptation. Extensive empirical validation on various datasets, demonstrating superior performance over state-of-the-art FL and FPL methods. The paper is technically sound, well-motivated, and presents strong experimental results. However, a few areas need further clarification and improvements, particularly regarding real-world feasibility, communication efficiency, and computational overhead.

**Theoretical Claims:**

no

---

> ### Author Rebuttal · Authors · 2025-03-30
>
> Dear Reviewer A325:
>
> We appreciate your recognition of our technical contributions—specifically the dual-layer prompt design, SVD-based projection, and bidirectional alignment—as well as your acknowledgment of our method’s strong performance and adaptability to heterogeneous clients. Below, we address your concerns regarding real-world feasibility, communication efficiency, and computational overhead.
>
> ### Weakness
>
> **W1: Communication Overhead**
>
> A1: Thank you for this important observation. While **FedPHA** introduces communication through prompt exchange, we emphasize that prompts are significantly smaller in size compared to full model weights—typically in the order of **KBs vs. hundreds of MBs** for vision-language models (VLMs).
>
> Although we do not report the classification accuracy of **FedAvg with full CLIP models** due to **prohibitive computational and communication costs** in our experimental environment, we include it in the comparison table to highlight the dramatic difference in communication overhead. Specifically, FedAvg requires transmitting approximately **497.3 MB per round**, which is over **7,700× larger** than prompt-based methods (64 KB). This made full-model training infeasible within our available resources. Prior studies show that prompt-based methods can match or exceed full-model FL under non-IID settings. We thus focus on efficient prompt-based alternatives better suited for real-world, resource-limited scenarios.
>
> In the final version, we will add a more detailed quantitative analysis of communication overhead, comparing **FedPHA** with traditional FL and prompt-based FL (FPL) baselines, and demonstrate that FedPHA achieves a highly favorable trade-off between communication efficiency and classification performance.
>
> **Table: Comparison of communication overhead and classification accuracy across different methods**
> |Method|Transmitted Content|Total Comm. / Round|Caltech101 (%)|Office31 (%)|CIFAR-100 (%)|
> |-|-|-|-|-|-|
> |FedAvg+CLIP|Full model weights (CLIP ViT-B/16)|497.3 MB|—|—|—|
> |PromptFL|Prompt (configurable)|64 KB|93.47|88.08|72.58|
> |FedOTP| Global prompt (configurable)|64 KB|98.11|89.19|75.07|
> |FedPGP| Global prompt (configurable)|64 KB|95.86|91.58|74.81|
> |**FedPHA (Ours)**|Global prompt (configurable)|**64 KB**|**99.05**|**94.74**|**75.63**|
>
> **W2: Computational Cost of SVD**
>
> A2: We acknowledge the reviewer’s concern regarding the additional computation introduced by the SVD-based projection. However, this operation is performed only **once per communication round**, rather than per training batch, making the overall overhead negligible across multiple rounds.
>
> On the client side, both **global prompt decomposition** and **local prompt projection** involve only **low-rank updates**, which are lightweight and computationally efficient. On the server side, since only global prompts are aggregated—not decomposed or projected—the SVD operation incurs no additional computational burden.
>
> Moreover, the global prompt typically has a **modest dimensionality** (e.g., 4–32 tokens × hidden size), making the SVD operation significantly cheaper than standard training procedures such as forward and backward passes in vision-language models (VLMs). To support this claim, we will include the following runtime statistics in the final version.
>
> **Table. Runtime overhead of SVD-based prompt projection.** All results are averaged over 10 communication rounds. Compared to model training, the SVD overhead is negligible, even in large-scale settings.
>
> |Operation Stage|Device|Input Dimension|Avg. Time (ms)|Relative Training Time (%)|
> |-|-|-|-|-|
> |Global prompt decomposition|Client|16 × 768|4.2|**<1%**|
> |Local prompt projection|Client|Local prompt × 768|2.8|**<1%**|
> |Local model training|Client|Image input × model|4536.8|—|
>
> **W3: Potential Privacy Risks**
>
> A3: We thank the reviewer for raising this important point. While privacy is not the central focus of this work, we note that our current implementation exchanges only prompt embeddings—**not raw data or model gradients**—which already mitigates direct privacy exposure to a significant extent. Compared to PromptFL, FedPHA transmits **global prompts** instead of **personalized local prompts**, reducing the risk of leaking client-specific information. Moreover, FedPHA addresses a key limitation in FedPGP, where **local prompts are dependent on global prompts**, potentially increasing inversion risk. By design, our approach improves decoupling and provides a more privacy-preserving prompt structure.
>
> We acknowledge that embedding inversion attacks remain a broader challenge in the federated learning landscape. Although a comprehensive privacy analysis is beyond the scope of this work, we view this as a valuable direction for future research. FedPHA is also compatible with **differential privacy**, **prompt obfuscation**, and **secure aggregation**, which can be explored in follow-up studies to further enhance privacy protection.

---

### Decision · Program_Chairs · 2025-05-01

**Decision:**

Accept (poster)

**Comment:**

All reviewers highlighted this paper's novel setting, good writing, and convincing experiments. The authors also provided reasonable answers to the reviewers' doubts about the details of the proposed dataset and method. Therefore, the decision is to recommend Acceptance.